# Square-root higher-order Weyl semimetals

Lingling Song ®[1], Huanhuan Yang ®[1], Yunshan Cao[1] & Peng Yan ®[1]✉

The mathematical foundation of quantum mechanics is built on linear algebra, while the application of nonlinear operators can lead to outstanding discoveries under some circumstances, such as the prediction of positron, a direct outcome of the Dirac equation which stems from the square-root of the Klein-Gordon equation. In this article, we propose a model of square-root higher-order Weyl semimetal (SHOWS) by inheriting features from its parent Hamiltonians. It is found that the SHOWS hosts both "Fermi-arc" surface and hinge states that respectively connect the projection of the Weyl points on the side surface and arris. We theoretically construct and experimentally observe the exotic SHOWS state in three-dimensional (3D) stacked electric circuits with honeycomb-kagome hybridizations and double-helix interlayer couplings. Our results open the door for realizing the square-root topology in 3D solid-state platforms.

Nearly all the operators encountered in quantum mechanics are linear (or antilinear) operators, such as the rotation, translation, parity, time reversal, etc, which allows us to construct the mathematical basis of quantum mechanics formulated on linear algebra. Square-root operator is one of the few exceptions. Historically, Paul Dirac derived the Dirac equation through a square-root operation on the Klein-Gordon (KG) equation to describe all spin-$\frac{1}{2}$ massive particles that inherit the Lorentz-covariance of the parent KG equation[1–3]. The approach has inspired Arkinstall et al.[4] to propose the concept of square-root topological insulator (TI) by taking the nontrivial square-root of a tight-binding (TB) Hamiltonian in periodic lattices. The most appealing feature of square-root TI is that it inherits the nontrivial nature of Bloch wave function from its parent Hamiltonian. The square-root TI was subsequently observed in a photonic cage[5]. Recently, the square-root operation has been applied to higher-order topological insulators (HOTIs) that allow topologically robust edge states with codimension larger than one[6–16]. Besides the gapped solution, e.g., the electron-positron pair, the Dirac equation allows another crucial gapless or massless solution called Weyl fermion[17] that plays an important role in quantum field theory and the standard Model. Although not yet observed among elementary particles, Weyl fermions are shown to exist as collective excitations in Weyl semimetals[18–21]. For the conventional Weyl semimetal, the three-dimensional (3D) topology features two-dimensional (2D) gapped surface states that connect the projection of Weyl points on the surface[18–21]. Very recently,

higher-order Weyl semimetal was reported which supports both the 2D surface Fermi arcs and the one-dimensional (1D) hinge state[22–29]. It is thus intriguing to ask if the square-root operation can apply to semimetals[30] or higher-order semimetals, and particularly how to realize these exotic states in experiments.

In this article, we propose a TB model of the square-root higher-order Weyl semimetal (SHOWS) by a vertical stacking of 2D square-root HOTIs with interlayer couplings in a double-helix fashion. It is found that the SHOWS hosts both 2D surface arc states and 1D hinge states with the topological feature being fully characterized by the quantized bulk polarization or edge invariant. We construct the TB model in stacked honeycomb-kagome (HK) hybridized inductor-capacitor (LC) circuit networks. By performing both the impedance and voltage measurements, we identify the fingerprint of the SHOWS by directly observing the Weyl points, the "Fermi arc" surface states, and the hinge states. It is revealed that both the surface states and the hinge states ideally connect the projections of the Weyl points on side surface and arris respectively, consistent with theoretical calculations.

## Results
### Model
Figure 1a shows the lattice structure of the proposed model, the square of which can be viewed as the direct sum of a stacked honeycomb and a breathing kagome lattices (Fig. 1b and the analysis in

---

[1]School of Electronic Science and Engineering and State Key Laboratory of Electronic Thin Films and Integrated Devices, University of Electronic Science and Technology of China, Chengdu 610054, China. ✉e-mail: yan@uestc.edu.cn

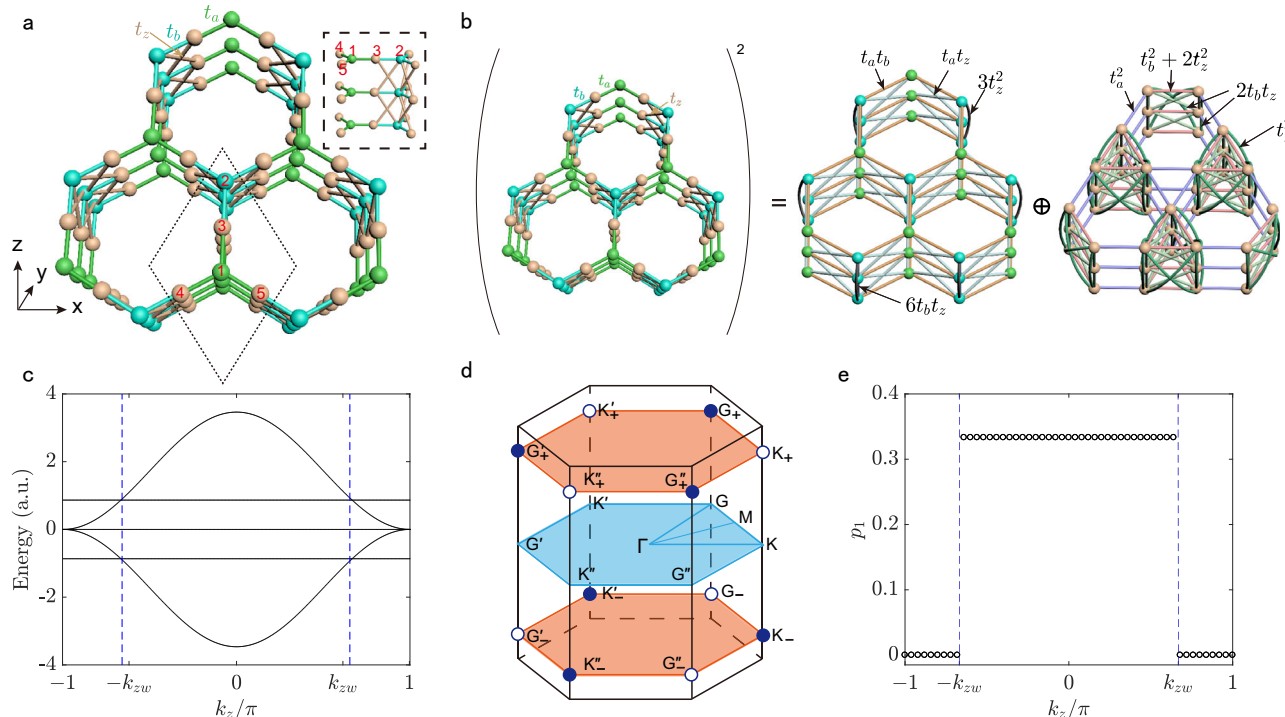

**Fig. 1 | 3D stacked HK TB model. a** Illustration of an infinite 3D stacked HK TB model. The unit cell including five nodes is represented by the dashed black rhombus. The intralayer hoppings are $t_a$ (green) and $t_b$ (blue) in the $x-y$ plane, whereas the interlayer double-helix hopping is $t_z$ (brown). Insets: Right view of the cell. **b** The equivalence between the squared Hamiltonian of the HK circuit and its parents. **c** The bulk dispersion along the $k_z$ direction with $(k_x, k_y) = (4\pi/3, 0)$. The dashed blue line indicates the position of the degenerate points. **d** The first Brillouin zone and the distribution of the Weyl points. The hallow and solid circles represent the Weyl point with the charge ± 1. **e** Bulk polarization $p_1$ as a function of $k_z$. For TB calculations in **c** and **e**, we set $t_a = 0.5$, $t_b = 1$, and $t_z = 0.5$.

Supplementary Note 1). The TB Hamiltonian is given by

$$
\begin{aligned}
\mathcal{H} = & \, t_a \sum_{\langle m,n \rangle} \left( a_m^\dagger c_n + a_m^\dagger d_n + a_m^\dagger e_n \right) \\
& + t_b \sum_{\langle m,n \rangle} \left( b_m^\dagger c_n + b_m^\dagger d_n + b_m^\dagger e_n \right) \\
& + t_z \sum_{\langle\langle m,n \rangle\rangle} \left( b_m^\dagger c_n + b_m^\dagger d_n + b_m^\dagger e_n \right) + \text{H.c.},
\end{aligned}
\tag{1}
$$

where $a^\dagger$ ($a$), $b^\dagger$ ($b$), $c^\dagger$ ($c$), $d^\dagger$ ($d$), and $e^\dagger$ ($e$) are the creation (annihilation) operators on the site 1-5, respectively, $\langle m, n \rangle$ and $\langle\langle m, n \rangle\rangle$ label the nearest-neighbor and next-nearest-neighbor coupling, respectively, and $t_a$, $t_b$, and $t_z$ are the hopping parameters. H.c. represents the Hermitian conjugate. In Fig. 1a, the nearest-neighbor sites of node 1(2) are nodes 3,4,5 with $t_a$, $t_b$ being the intralayer hopping parameters. The next-nearest-neighbor sites of node 2 are nodes 2, 3 and 4 in the adjacent layer with $t_z$ being the interlayer hopping parameter. Without loss of generality, we assume all hopping paramaters are positive. In momentum space, the Hamiltonian can be expressed as

$$
\mathcal{H} = \begin{pmatrix} O_{2,2} & \Phi_{\mathbf{k}}^\dagger \\ \Phi_{\mathbf{k}} & O_{3,3} \end{pmatrix},
\tag{2}
$$

where $O_{2,2}$ and $O_{3,3}$ are the $2 \times 2$ and $3 \times 3$ zero matrix, respectively, and $\Phi_{\mathbf{k}}$ is the $3 \times 2$ matrix

$$
\Phi_{\mathbf{k}} = \begin{pmatrix} t_a & t_b + 2t_z \cos(\mathbf{k} \cdot \mathbf{a}_3) \\ t_a & [t_b + 2t_z \cos(\mathbf{k} \cdot \mathbf{a}_3)] e^{-i\mathbf{k} \cdot \mathbf{a}_1} \\ t_a & [t_b + 2t_z \cos(\mathbf{k} \cdot \mathbf{a}_3)] e^{-i\mathbf{k} \cdot \mathbf{a}_2} \end{pmatrix}.
\tag{3}
$$

Here $\mathbf{k} = (k_x, k_y, k_z)$ is the wave vector, and $\mathbf{a}_1 = \frac{1}{2}\hat{x} + \frac{\sqrt{3}}{2}\hat{y}$, $\mathbf{a}_2 = -\frac{1}{2}\hat{x} + \frac{\sqrt{3}}{2}\hat{y}$ and $\mathbf{a}_3 = \hat{z}$ are three basic vectors.

By taking the square of the original Hamiltonian (2), we can conveniently obtain the dispersion relation of $[\mathcal{H}]^2$ (see Supplementary Note 1)

$$
E_{\mathbf{k}} = 0 \quad \text{and} \quad \frac{3}{2} \left[ t_a^2 + t_b'^2 \pm \sqrt{(t_a^2 - t_b'^2)^2 + 4t_a^2 t_b'^2 |\Delta(\mathbf{k})|^2} \right],
\tag{4}
$$

with $t_b' = t_b + 2t_z \cos(k_z)$ and $\Delta(\mathbf{k}) = (1 + e^{i\mathbf{k}\cdot\mathbf{a}_1} + e^{i\mathbf{k}\cdot\mathbf{a}_2})/3$. The band structure of the original Hamiltonian is thus given by $\varepsilon_{\mathbf{k}} = \pm\sqrt{E_{\mathbf{k}}}$. It is found that the band structure closes at the twofold degenerate points $K_\pm = (4\pi/3, 0, \pm k_{zw})$, as shown in Fig. 1c, with $k_{zw} = \arccos[(t_a - t_b)/(2t_z)]$ when $|t_a - t_b| < 2t_z$. It is straightforward to verify that their time-reversal counterparts are $G'_\pm = (-4\pi/3, 0, \pm k_{zw})$, and their equivalence points locate at $G_\pm$, $G''_\pm$, $K'_\pm$, and $K''_\pm$, as shown in Fig. 1d. By evaluating the topological charge $C_{\text{FS}}$, we find that the hollow and solid circles plotted in Fig. 1d denote the Weyl points with opposite topological charges, i.e., $+1$ and $-1$, respectively (see Supplementary Fig. 1). In addition, we derive the low-energy effective Hamiltonian near the degeneracy points, and obtain a linear crossing in the vicinity of the Weyl points (Supplementary Note 2). The computation of Berry curvatures are plotted in Supplementary Fig. 1c, d, which indeed demonstrates that the Weyl points manifest as singularities (source and drain), a close analog to the magnetic monopole in momentum space.

For a system with the rotational symmetry (it is $C_3$ in our model), the bulk polarization is the appropriate invariant to characterize the topological features. For the $n$th band, the bulk polarization[7] as a

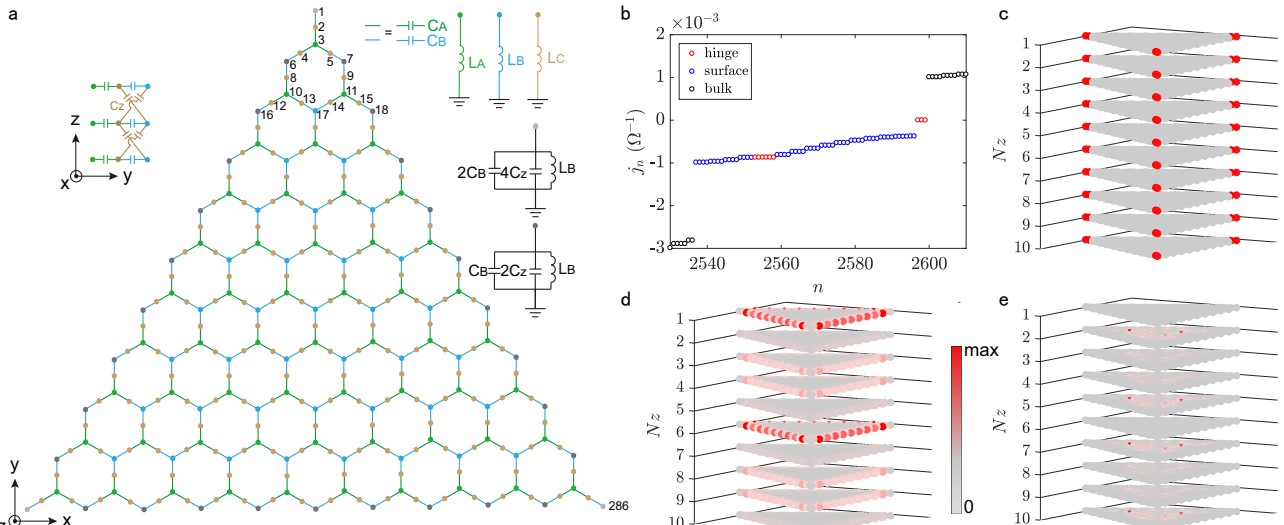

**Fig. 2 | Finite-size circuit model. a** Top view of a stacked 10-layer HK circuit with 2860 nodes. The green and blue segments represent the capacitors $C_A$ and $C_B$, respectively, and each node is grounded by capacitors and inductors with the configurations shown in the insets. The finite-size equilateral triangular structure with the zigzag edge was chosen because one requires a uniform onsite admittance for all nodes of the breathing kagome block that appears upon squaring the Laplacian. **b** Admittances for $C_A = C_Z = C_B/2 = 0.5$ nF, $L_A = 30$ μH, $L_B = 7.5$ μH, $L_C = 18$ μH, and $L_G = 21.829$ μH. The red, blue, black symbols represent the hinge, surface, bulk states, respectively. Spatial distribution of the wave functions of the normalized hinge state ($j_n = 3.534 \times 10^{-7}$ $\Omega^{-1}$) (**c**), surface state ($j_n = -0.0003762$ $\Omega^{-1}$) (**d**), and bulk state ($j_n = 0.001042$ $\Omega^{-1}$) (**e**).

function of $k_z$ is written as

$$2\pi p_n(k_z) = \arg \theta_n(\mathbf{k}) \,(\mathrm{mod}\ 2\pi), \tag{5}$$

where $\mathbf{k} = (4\pi/3, 0, k_z)$, and $\theta_n(\mathbf{k}) = u_n^\dagger(\mathbf{k}) U_\mathbf{k} u_n(\mathbf{k})$ with $u_n(\mathbf{k})$ the $n$th eigenvector and the $U$-matrix

$$U_\mathbf{k} = \begin{pmatrix} 1 & 0 & 0 & 0 & 0 \\ 0 & e^{-i\mathbf{k}\cdot\mathbf{a}_2} & 0 & 0 & 0 \\ 0 & 0 & 0 & 0 & 1 \\ 0 & 0 & 1 & 0 & 0 \\ 0 & 0 & 0 & 1 & 0 \end{pmatrix}. \tag{6}$$

Here we are particularly interested in the 1st (or 5th) band, because the Weyl points only appear in the intersecting between the first and second energy bands (or between the fourth and fifth energy bands). As shown in Fig. 1e, $p_1$ takes 1/3 for $|k_z| < |k_{zw}|$, and 0 for $|k_z| > |k_{zw}|$. The topological phase transition occurs at $k_z = \pm k_{zw}$. A non-vanishing $p_1$ indicates the very presence of the higher-order topologial states. As a comparison, we also calculate the edge topological invariant following the method of[10,31]. We identify the same phase transition point at $\pm k_{zw}$, as shown in Supplementary Fig. 3b. The present model unambiguously demonstrates the bulk-hinge correspondence and manifests itself as an ideal SHOWS (Supplementary Fig. 2d). It is noted that a pair of Weyl points emerge with opposite wave vectors (Fig. 1c) because of the inversion-symmetry breaking in our model. It is worth mentioning that the present model also allows a 3D square-root HOTI phase (Supplementary Fig. 2e–g). Electronic circuits are an excellent platform to study topological physics[32–43]. In what follows, we construct the TB SHOWS model in 3D stacked HK $LC$ circuits.

## Circuit realization of SHOWS

We consider a stacked 10-layer HK circuit with $\mathcal{N} = 2860$ nodes, as depicted in Fig. 2a. The dots and lines represent nodes and capacitors, respectively. The circuit dynamics at frequency $\omega$ obeys Kirchhoff's law $I_a(\omega) = \sum_b J_{ab}(\omega) V_b(\omega)$, with $I_a$ the external

current flowing into node $a$, $V_b$ the voltage of node $b$, and $J_{ab}(\omega)$ being the circuit Laplacian

$$J(\omega) = \begin{pmatrix} J_{0B} & -J_B & 0 & \dots & 0 & -J_Z & 0 & \cdots_1 \\ -J_B & J_{0C} & -J_A & \cdots & -J_Z & 0 & 0 & \cdots_2 \\ 0 & -J_A & J_{0A} & \cdots & 0 & 0 & 0 & \cdots_3 \\ \vdots & \vdots & \vdots & \ddots & \vdots & \vdots & \vdots & \ddots \\ 0 & -J_Z & 0 & \cdots & J_{0B} & -J_B & 0 & \cdots_{287} \\ -J_Z & 0 & 0 & \cdots & -J_B & J_{0C} & -J_A & \cdots_{288} \\ 0 & 0 & 0 & \cdots & 0 & -J_A & J_{0A} & \cdots_{289} \\ \vdots_1 & \vdots_2 & \vdots_3 & \vdots & \vdots_{287} & \vdots_{288} & \vdots_{289} & \ddots \end{pmatrix} \tag{7}$$

with $J_{0A} = 3i\omega C_A - 1/(i\omega L_A)$, $J_{0B} = 3i\omega(C_B + 2C_Z) - 1/(i\omega L_B)$, $J_{0C} = i\omega(C_A + C_B + 2C_Z) - 1/(i\omega L_C)$, $J_A = i\omega C_A$, $J_B = i\omega C_B$, $J_Z = i\omega C_Z$ and the subscript of the dots indicating the column/row numbers. Under the resonance condition $\omega_0 = 1/\sqrt{3C_A L_A} = 1/\sqrt{(3C_B + 6C_Z)L_B} = 1/\sqrt{(C_A + C_B + 2C_Z)L_C}$, the circuit Laplacian (7) exactly recovers the TB Hamiltonian by the following one-to-one correspondence: $-\omega_0 C_A \leftrightarrow t_a$, $-\omega_0 C_B \leftrightarrow t_b$, and $-\omega_0 C_Z \leftrightarrow t_z$. To explore the square-root topological semimetal phase, we set $C_A = C_B/2 = 0.5$ nF, $C_Z = 0.5$ nF and $L_A = 30$ μH, $L_B = 7.5$ μH, and $L_C = 18$ μH in the following calculations, if not stated otherwise.

To facilitate the detection of the hinge states through a direct two-point impedance measurement[39], we connect a grounded inductor $L_G = 22$ μH to all nodes to move the hinge modes to the zero admittance without modifying their wave functions[6]. By measuring the impedance, one can precisely characterize the wave function of the zero-admittance hinge states in circuit[6,39]. Figure 2b exhibits the corresponding admittance spectrum, where the red, blue, and black dots represent the hinge, surface, and bulk states, respectively. One can see the three-fold degeneracy of the in-gap hinge states due to the $C_3$ symmetry and generalized chiral symmetry[6]. In addition, the time reversal symmetry dictates the bulk/surface states being singlet or double-degenerate. The spatial distributions of each mode are plotted in Fig. 2c–e, from which one can straightforwardly distinguish them.

The photograph of 3D $LC$ electric circuits fabricated on a printed circuit board (PCB) is displayed in Methods. Our circuit is designed

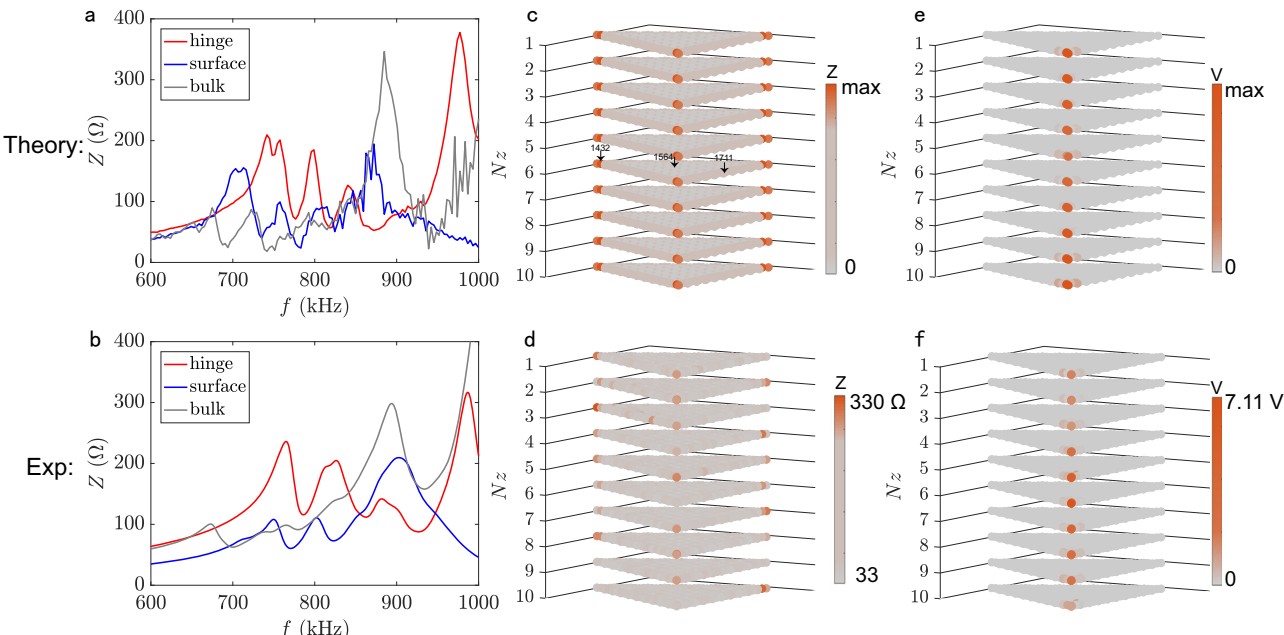

**Fig. 3 | Impedances spectra, impedances and voltage distributions.**
**a** Theoretical impedance versus the driving frequency in a disordered circuit.
**b** Measured impedance as a function of the frequency. Calculated (**c**) and measured
(**d**) impedance distribution of hinge state over the system. Hinge-state voltage
distribution in theory (**e**) and experiment (**f**).

by Altium Designer. The connection between different layers of the circuit boards is through flexible flat cable. We choose electric elements $C_A = C_B/2 = 0.5$ nF, $C_Z = 0.5$ nF and $L_A = 30$ μH, $L_B = 7.5$ μH, $L_C = 18$ μH and $L_G = 22$ μH, the same as those for theoretical computations above, but with a practical 2% tolerance. The resonant frequency is then $f_c = 1/(2\pi\sqrt{3C_A L_A}) = 755$ kHz. We first measure the impedance between three representative nodes and the ground as a function of the exciting frequency with the impedance analyzer (Keysight E4990A). Experimental results are shown in Fig. 3b, which agree well with theoretical calculations plotted in Fig. 3a. Here we select the 1432nd, 1564th, and 1711st nodes (specified in Fig. 3c) to characterize the properties of the hinge, surface, and bulk states, respectively. We then measure the spatial distribution of the impedance and voltage over the circuit (Fig. 3d, f), which compare reasonably well with the theoretical results plotted in Fig. 3c, e.

To characterize the hinge states more carefully, we project the dispersion to the $k_z$ axis, as shown by the color map in Fig. 4a. In theoretical calculations, one can conveniently set different admittances $j_n$ and analyze the spectrum. For circuit experiments, we can shift an arbitrary admittance to zero by adjusting the value of $L_G$. This is what we have done for measuring the hinge states. To demonstrate another experimental technique, we, alternatively, tune the frequency to measure the Fermi-arc surface state with the same circuit structure. Fortunately, by mapping Kirchhoff's law to the Schrödinger equation in circuit[40,41], we obtain the frequency dispersion (see Supplementary Note 5) that significantly facilitates our experimental measurements. A continuous variation of the frequency can be seen as an adiabatic transformation of our model, with everything else kept constant. Experimentally, we impose a voltage source in the middle of one hinge of the circuits, and scan the voltage distribution along the hinge. Specifically, we input a signal $v_s(t) = 5\sin(\omega t)$ V at a hinge node with the arbitrary function generator (GW AFG-3022), and then collect the voltage $v(\omega, z)$ with frequency $f = \omega/(2\pi)$ ranging from 500 kHz to 1600 kHz by using the oscilloscope (Keysight MSOX3024A). We perform the Fourier transformation on the $v(\omega, z)$ and obtain the projected dispersion along the $k_z$ direction, shown by the color map in Fig. 4b. It can be seen that the hinge states connect

two Weyl points at a resonant frequency around 860 kHz and 1441 kHz, which perfectly agrees with the simulation results marked by the solid red circles.

Furthermore, it is known that the "Fermi arc" surface state is an unique feature of Weyl semimetals. The Weyl points emerge at $j_n = \pm 0.004082$ $\Omega^{-1}$ (shown in Fig. 4a), which corresponds to frequencies $f = 835$ and 1441 kHz in Fig. 4b. However, $f = 1441$ kHz deviates a lot from the working frequency of our selected electric components, because the component values suffer a drastic change (the values of electric components depend on the frequency). So, we only consider $f = 835$ kHz to display the complete "Fermi arc" (Fig. 4d). Figure 4c shows the "Fermi arc" surface dispersion at $j_n = 0.004082$ $\Omega^{-1}$. Figure 4d shows the "Fermi arc" surface dispersion at $f = 835$ kHz. Here, we determine $k_x$ by $k_x = 2\pi m/N$, with $m = 1, 2, 3, \ldots N-1$, and $N$ being the number of grid points in the $\hat{x}$ direction. The geometry used for the experimental measurement in Fig. 4d is a side face of the triple prism. We consider periodic boundary condition along $\hat{z}$ direction and open boundary condition along both $\hat{x}$ and $\hat{y}$ directions. The color map represents the measured data and the white circles denote the simulated equal-admittance contour, whereas the hollow and solid dots denote the projections of Weyl points with opposite topological charges $+1$ and $-1$, respectively. Our experiment therefore unambiguously supports the bulk-hinge correspondence and identifies the emergence of SHOWS.

## Discussion

To summarize, we proposed a TB model of the SHOWS and constructed it in 3D double-helix stacked $LC$ circuits. Through the impedance and voltage measurements, we directly observed both the 1D prismatic states and the 2D "Fermi arc" surface states connecting the projection of Weyl points on the arris and side surface respectively, the fingerprint of SHOWS. Comparing with the normal Weyl semimetal[18], the SHOWS supports robust hinge states, besides the arc surface states. The emergence of Weyl pairs in SHOWS with both positive and negative energies marks its difference from the conventional higher-order Weyl semimetals[22–29]. One of the parent sublattices, i.e., the honeycomb lattice, originally does not support any hinge states or flat-

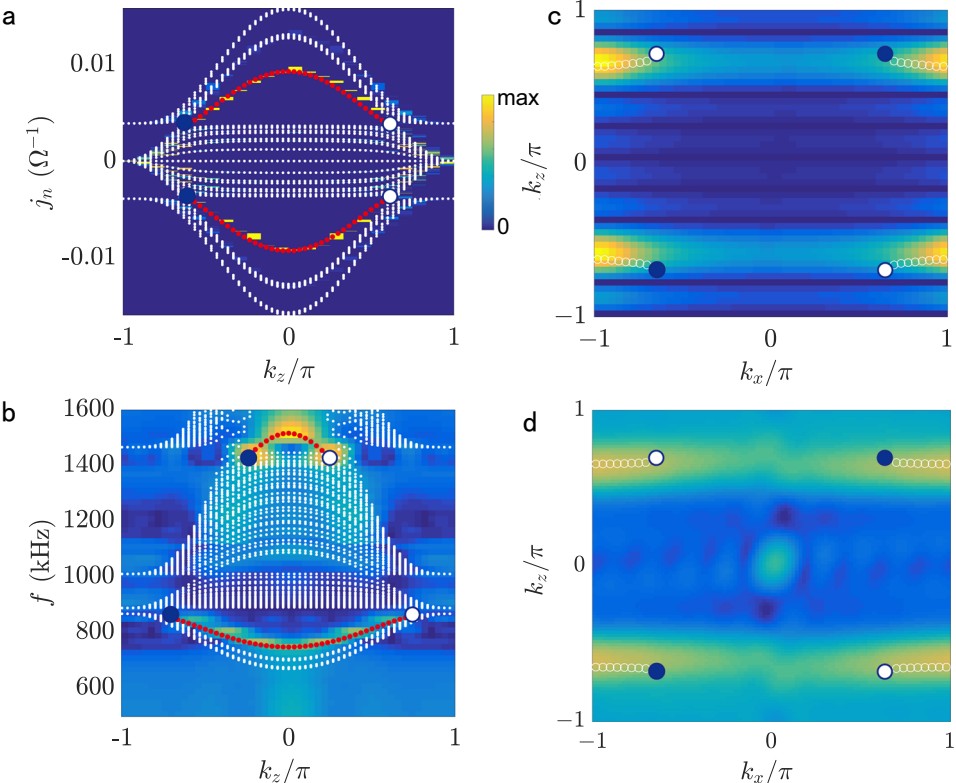

**Fig. 4 | Hinge states and Fermi arcs. a** The projected admittances along $k_z$ axis. **b** Hinge-state dispersion connecting the projection of the Weyl points on $k_z$ axis. The red dots and the color map in **a** and **b** represent the theory and experimental hinge spectrum, respectively. **c** The numerical "Fermi arc" of the surface states at $j_n = 0.004082\,\Omega^{-1}$, connecting the projections of the Weyl points on $k_x$-$k_z$ surface. **d** "Fermi arc" of the surface states at 835 kHz. The color map and the white circles represent the experimental and theoretical results, respectively.

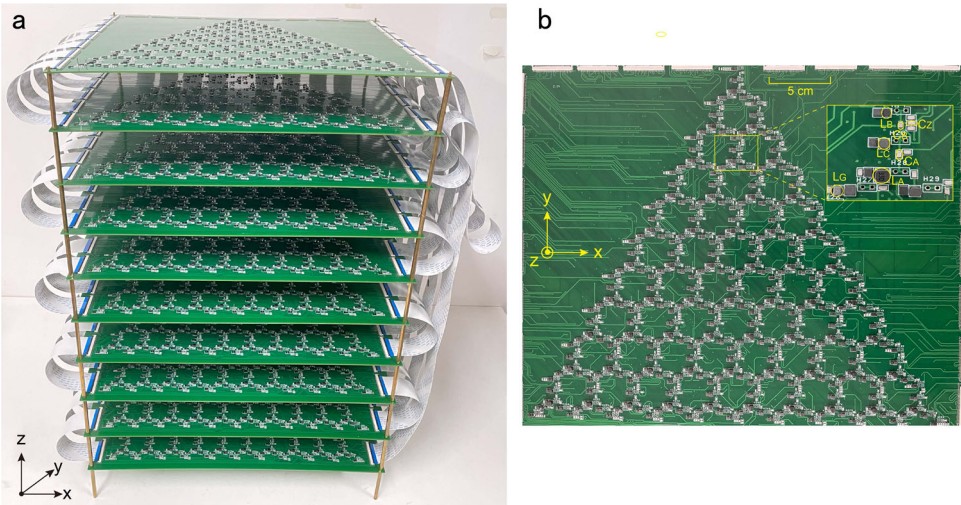

**Fig. 5 | The printed circuit board.** Side **a** and top **b** view of the printed circuit board in experiment.

band states. The square-root operator, however, makes it inherit these exotic states from the other parent sublattice. When the square-root operation is applied, the spectra of the blocks are necessarily degenerate[44] and the degenerate boundary modes of the honeycomb lattice are not topological but impurity states, stemming from onsite energy offsets at the boundary nodes[11]. In the present model, by adjusting the interlayer hopping $t_z$ to be breathing, we envision the emergence of third-order corner states[45].

From the application perspective, both hinge states and surface states can be used to design robust solid-state devices. For

example, one can transmit the signal with extremely low loss[46] and devise the multiplexing[47] and imaging[48] with these topologically boundary modes. Recently, it is shown that the topological LC circuit can be fully integrated by complementary metal-oxide-semiconductor (CMOS) technology[49,50], which may solve the outstanding challenges met in the chip industry. Our findings may stimulate the effort in the broad physics and materials community to look for real materials that support SHOWS, since we have provided a rather general framework to construct the Hamiltonian. It can be utilized to conduct electric charge with faster velocity[17,51]. Since Weyl particles are

**Table 1 | Electric elements used in experiments**

|  | Electric elements | Mean value | Tolerance |
|---|---|---|---|
| $C_A$ | FH 0805 (50 V) | 0.5 nF | ± 2% |
| $C_B$ | Walsin 0805 (50 V) | 1 nF | ± 2% |
| $C_Z$ | FH 0805 (50 V) | 0.5 nF | ± 2% |
| $L_A$ | SXN 4040 (1.1 A) | 30 μH | ± 2% |
| $L_B$ | Cjiang 2520 (680 mA) | 7.5 μH | ± 2% |
| $L_C$ | Cjiang 3030 (560 mA) | 18 μH | ± 2% |
| $L_G$ | FH 2521 (380 mA) | 22 μH | ± 2% |

The capacitors are produced by FH and Walsin company, with the 0805 package. The inductors belong to the low-resistance power surface-mount devices, produced by SXN, Cjiang, and FH company, with the 4040, 2520, 3030, and 2521 package. The tolerance of circuit elements are selected within 2% manually.

topologically protected from scattering[19,20], they could be useful in quantum information science. One may use the square-root method to explore many interesting new ideas for further study. For example, in electrical circuits, one may discover the enchanting phenomenon with square-root operations in higher dimensions, e.g., four-dimension (4D)[52]. One may apply square-root operation to the Floquet TIs or semimetals, sustained by time-dependent periodic Hamiltonians[50,53]. Beyond the square-root, one can generalize the approach to the $2^n$-root weak, Chern, and higher-order topological insulators, and $2^n$-root topological semimetals[11,15]. Our results thus pave the way to realizing the square-root higher-order topological states in electric circuits, and may inspire the exploration in other solid-state systems, such as cold atoms, photonic crystals, and elastic lattices.

## Methods

### Circuit Laplacians and PCB images in experiments

The circuit dynamics at frequency $\omega$ obeys Kirchhoff's law $I_a(\omega) = \sum_b J_{ab}(\omega)V_b(\omega)$, with $I_a$ the external current flowing into node $a$, $V_b$ the voltage of node $b$, and $J_{ab}(\omega)$ being the circuit Laplacian

$$J_{ab}(\omega) = i\mathcal{H}_{ab}(\omega) = i\omega\left[-C_{ab} + \delta_{ab}\left(\sum_n C_{an} - \frac{1}{\omega^2 L_a}\right)\right], \quad (8)$$

where $C_{ab}$ is the capacitance between $a$ and $b$ nodes, $L_a$ is the grounding inductance of node $a$, and the sum is taken over all nearest-neighboring nodes. For a finite circuit, the Laplacian of the circuit can be written as Eq. 7 in the main text. Considering the resonance condition $\omega = \omega_0$, one can obtain all eigenvalues $j_n$ (admittances) and eigenfunctions $\psi_n$ ($n = 1, 2, \ldots, \mathcal{N}$). The geometry used for the experimental is a triangular prism shown in Fig. 5a. The geometry used for the theoretical calculations in Fig. 4a, b is also a triangular prism (open boundary condition in $k_x - k_y$ plane and periodic boundary condition along $k_z$ direction) including 286 nodes in each layer. The geometry used for the theoretical results in Fig. 4c, d, Supplementary Fig. 3a, Supplementary Fig. 4 and Supplementary Fig. 5 is a slab (open boundary condition along $k_y$ direction and periodic boundary condition along both $\hat{x}$ and $\hat{z}$ directions) containing 155 nodes.

Table 1 lists electric elements used in experiments.

In the calculation of Fig. 2 in the main text, we consider the ideal situation that all inductors and capacitors have no loss and disorder. Considering the practical loss and tolerance of capacitors and inductors, we introduced 2% disorder to each capacitor and inductor in theoretical calculations thereafter. In experiments, we stacked ten identical 2D printed circuit boards (PCBs) along the $\hat{z}$ direction, as shown in Fig. 5a. Figure 5b shows the top view of the PCB with the inset zooming in the design details of the electrical circuit.

## Data availability

The data that support the plots within this paper and other findings of this study have been deposited in the Zenodo database : https://doi.org/10.5281/zenodo.6976420.

## Code availability

We used the commercial software MATLAB to perform the numerical calculations. Requests for the computation details can be addressed to the corresponding author.

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

## Acknowledgements

We acknowledge Z.-X. Li for useful discussions. This work was supported by the National Natural Science Foundation of China (Grants No. 12074057. P.Y., No. 11704060. Y.C. and No. 11604041. P.Y.).

## Author contributions

P.Y. conceived the idea and contributed to the project design. L.S. and H.Y. designed the circuits and performed the measurements. L.S. developed the theory and wrote the manuscript. L.S., H.Y., Y.C. and P.Y. discussed the results and revised the manuscript.

## Competing interests

The authors declare no competing interests.
