## [Peer Review File · Nature Communications]

REVIEWER COMMENTS

Reviewer #1 (Remarks to the Author):

The current manuscript presents a lattice structure that host some topological features related to higher order Weyl semimetals. The authors establish the relation between the proposed lattice and that formed by taking the square of the Hamiltonian in the momentum representation and they relate this to previous work on square root topological insulators. Finally, the presented lattice structure is physically implemented by using electric circuits.

In my opinion, the manuscript is not suitable for publication in its current format. In the following, I explain in details some of the problems with this work:

1. The manuscript is poorly written in terms of its English language even at the abstract level. For instance, the first sentence in the abstract seems to be missing an appropriate ending. In the third sentence of the abstract, the authors states "... the projection of the Weyl point". It is not clear projection on what? And so on throughout the manuscript.
2. While the authors emphasize the role of taking the square root in order to construct topological lattices, this is not what they do here. If I understand correct, the start with a topological lattice and use it as a starting point to construct another lattice. The square root operation (more accurately square operation) is only used to facilitate the analysis.
3. The correspondence between the tight binding Hamiltonian in Eq.(1) and Fig.1(a) is not clear. I am not suggesting it is wrong, I am just saying that by looking at the Hamiltonian and the figure, it is difficult to relate the relevant terms to the corresponding sites. A better, enlarged figure with more appropriate description is needed.
4. Similarly, it is difficult to understand the actual circuit structure from Fig.2 (a). I could not understand what the dots represent and how to relate them to the actual electric elements.
5. I could not understand how the features of the presented structure actually relate to the Weyl-semimetal.
6. Also, I could understand what is the big picture here? Why this work is important? Does it present any new physics? Does it have any interesting application? Does it provide a solution to any open problem? Does it provide more insight into a specific problem? Can it pave the way for exploring more interesting ideas? There is no discussion whatsoever about the impact or importance.

In summary, I believe that the manuscript needs to be improved dramatically before it can be considered for publication in general. In addition, when it comes to publication in Nature Communications, the authors need to clarify what is the impact and significance of their work before it can be considered.

Reviewer #2 (Remarks to the Author):

The authors study a square-root higher-order Weyl semimetal by generalizing to 3D in a specific way (namely one that breaks M_y symmetry, and therefore I -symmetry, for $t_z > 0$, even when $t_a = t_b$) the 2D model considered in their previous work (Ref. [6]), and implemented it in an electrical circuit. Due to the square-root nature of the model, the Weyl pairs appear at symmetric energies, which is a novel feature brought forth by this study. The experimental realization shows unambiguous signatures of the Weyl points, hinge states and Fermi arcs of the system. Considering the recent surge of interest in square-root topology and its generalizations, as well as in Weyl semimetals, I believe these results will appeal to a broad community.

While I find these results interesting and view them in a favorable light, their presentation feels a bit rushed and insufficiently revised by the authors before submission and, as a consequence, there are several comments, listed below, that have to be adequately addressed first by the authors before I can recommend this paper for publication.

(1) A reference to Eq. (5) should be included, possibly Ref. [7] again. In lines 36-39, 94-96 and 107-110, the authors connect the topology of the hinge states to the quantized bulk polarization. However, it has been recently shown that, in itself, this quantization is not sufficient to affirm the topological nature of these states – see, e.g., npj Quantum Materials 5, 63 (2020) and the authors' Ref. [10] (which should be corrected, since it was published in PRB), where the zero-energy corner states of the breathing kagome lattice (the parent block of the authors' model at each k_z) are shown to stem from edge invariants and not bulk invariants. As such, the lines mentioned above should be less assertive when relating the topological nature of the hinge states to the bulk polarization, and also a reference to the results of Ref. [10] should be made.

(2) In Fig. 2(a), it should be explained why the bottom line of nodes is included, since they belong to incomplete unit cells (the reason is that one requires a uniform onsite admittance for all nodes of the breathing kagome block that appears upon squaring the Laplacian). Also, the bottom edge line of blue nodes should be colored in dark gray instead.

(3) In Fig. 2(b), while the hinge states are understandably 3-fold degenerate, I would expect the same to hold for the surface and bulk states, which appear to be instead 4-fold degenerate (or quasi-degenerate). Is there a simple reason for this?

(4) The parameters defined below the Laplacian in eq. (7) are not correctly defined: (i) $J_{\{0B\}}$ should read as $J_{\{0B\}}=iw(3C_B+6C_z)-1/(iwL_B)$ instead, (ii) similarly $J_{\{0C\}}$ should read as $J_{\{0C\}}=iw(C_A+C_B+2C_z)-1/(iwL_C)$, and (iii) J_B is simply $J_B=iwC_B$, with another parameter, let us call it $J_Z=iwC_Z$, connecting nodes at adjacent layers. Notice that all diagonal inductor terms, including in $J_{\{0A\}}$, should have a negative sign.

(5) In lines 163-164, is there a way for the authors to specify (e.g. with arrows) in one of the plots where the nodes selected for the impedance measurements are located?

(6) Concerning Fig. 4(b), I understand that the frequency spectrum was found through the method of Sec. V of the Supp. Inf. However, given that the resonance condition occurs at $f_c=755\text{kHz}$, for other values of "f" the model will not be exactly the same (there are varying onsite admittances at nodes A, B and C away from f_c). Then $f=860\text{kHz}$ is considered in Figs. 4(c) and 4(d), such that it corresponds to a slightly different model. Even though I fully appreciate the relevance of the results of Fig. 4, is there an argument – based, e.g., on an adiabatic connection to the $f_c=755\text{kHz}$ case – from where one can say that qualitatively similar results are expected for this case?

(7) Still on Figs. 4(c) and 4(d), what is the origin of the central peak in Fig. 4(d)? Also, how is k_x determined since there are OBC along x? It is not clear what the exact geometry used for the experimental results in Fig. 4(d) is, since it seems to assume PBC along x and z but OBC along y. The authors should explain this figure in more detail.

(8) In the Methods section, the specific components used should be identified, as well as their tolerances (is it 2% for all of them, was it measured?).

(9) In lines 216-221, the authors relate the boundary modes of the residual block to those of the parent block of the squared Hamiltonian. I see these blocks on the same level, such that it might not be completely precise to say the boundary modes of the residual block are inherited from parent block. This should be more clearly restated, e.g., by citing Phys. Rev. Research 2, 033397 (2020) and saying, as in this reference,

that the spectra of the blocks are necessarily degenerate, apart from extra zero-energy bands at the residual block, while explaining at the same time that the degenerate boundary modes of the residual lattice are not topological but impurity states, stemming from onsite admittance offsets at the boundary nodes when the squaring operation is applied, as illustrated in Fig. 9(a) of Ref. [11].

(10) When OBC are considered along all directions, are third-order corner states present, albeit possibly buried in the bulk continuum? Is there a simple strategy or modification to the model that could make these states appear at a gap?

(11) In Figs. S1(c) and (d), should the caption not say instead something like "Berry curvature of the lower band of h_k^H for (c) $k_z=k_{zw}$ and (d) $k_y=\pm 2\pi/\sqrt{3}$ "? Otherwise, it is not easy to understand what are the regions being considered, as they are not only around K^+ and K^- as indicated there and below Eq. (S10).

(12) In Fig. S2, the parameters in a-c should be swapped with those in d-f, since the $|t_a-t_b|<2t_z$ has to be met for the Weyl points to appear. Also in Fig. S2(a), can the authors indicate to what surfaces the K and \bar{K} points belong to in the BZ? Comparing with Fig. 1(c), it would seem K is not simply $(4\pi/3,0,0)$, but belongs instead to a surface with $-\pi<k_z<k_{zw}$, while \bar{K} belongs to a symmetric $k_{zw}<k_z<\pi$.

(13) There are three papers in a pre-print version on square-root topology that I recommend the authors either cite as is or monitor and wait for their publication in the meantime before citing them:

arXiv:2108.01366 (2021)

arXiv:2109.00879 (2021)

arXiv:2111.14327 (2021)

Misprints:

- Line 9: inherent → inherit
- Line 154: $L_A=33 \mu\text{H}$ → $L_A=30 \mu\text{H}$
- Below Eq. (S1): matrixs → matrices

Reply to reviewers' reports on Nature Communications manuscript NCOMMS-22-04759 by Song *et al.*

Response to Reviewer 1:

Comment 1: The current manuscript presents a lattice structure that host some topological features related to higher order Weyl semimetals. The authors establish the relation between the proposed lattice and that formed by taking the square of the Hamiltonian in the momentum representation and they relate this to previous work on square root topological insulators. Finally, the presented lattice structure is physically implemented by using electric circuits. In my opinion, the manuscript is not suitable for publication in its current format. In the following, I explain in details some of the problems with this work:

Reply: We thank the reviewer for spending his/her time to review our work with useful comments. To address the concerns from our respected reviewer, we have revised our manuscript carefully to improve the readability and to emphasize the significance and impact of our work.

Comment 2: The manuscript is poorly written in terms of its English language even at the abstract level. For instance, the first sentence in the abstract seems to be missing an appropriate ending. In the third sentence of the abstract, the authors states "... the projection of the Weyl point". It is not clear projection on what? And so on throughout the manuscript.

Reply: We appreciate the reviewer's criticism. We have carefully polished the language over the whole manuscript:

The first sentence in the abstract was revised as "*The mathematical foundation of quantum mechanics is built on linear algebra, while the application of nonlinear operators can lead to outstanding discoveries under some circumstances, such as the prediction of positron, a direct outcome of the Dirac equation which stems from the square-root of the Klein-Gordon equation.*"

The third sentence in the abstract was modified as "*The SHOWS hosts both "Fermi-arc" surface state and hinge state that respectively connect the projection of the Weyl points on the side surface and arris*".

The last sentence in the introduction was corrected as "*It is revealed that both the surface states and the hinge states ideally connect the projections of the Weyl points on the side surface and arris, respectively, consistent with theoretical calculations.*"

The caption in Fig. 4 was revised as "*Hinge state dispersion connecting the projections of the Weyl points on axis.*" and "*The numerical "Fermi arc" of the surface states at $= 0.004082^{-1}$ connecting the projection of the Weyl points on surface.*" The second sentence in Discussion and Conclusion was modified as "*Through the impedance and voltage measurements, we directly observed both the 1D prismatic states and the 2D "Fermi-arc" surface states connecting the projection of Weyl points on the arris and side surface respectively, the fingerprint of SHOWS.*"

All corrections have been highlighted in the revised manuscript.

Comment 3: While the authors emphasize the role of taking the square root in order to construct topological lattices, this is not what they do here. If I understand correct, the start with a topological lattice and use it as a starting point to construct another lattice. The square root operation (more accurately square operation) is only used to facilitate

the analysis.

Reply: We thank the reviewer for the critical comments. The square root operation indeed significantly simplifies our analysis. In the present study, it also provides ideas for us to construct new lattices and to realize novel topological states. As shown in Fig. R1, it is found that by applying the square-root operators to conventional topological insulators, one can construct a series of new topological phases.

Fig. R1 Square-root topological insulator [4].

Very recently, the square-root approach was applied to high-order topological insulators (shown in Fig. R2).

Fig. R2 Square-root higher-order topological insulator [6].

These results show that the square-root operation brings us many new topological structures and topological states.

Fig. R3 Square-root higher-order Weyl semimetals (this work) constructing by “square-root” of combined honeycomb and Kagome lattices.

In the present article, we apply the square-root operation to a three-dimensional gapless system and construct higher-order topological semimetals by mixing a 3D honeycomb lattice with the 3D breathing Kagome lattice [27], see Fig. R3. This approach may set

a paradigm for realizing the square-root higher-order topology in other solid-state systems, such as optical, acoustic, and plasmonic platforms.

Comment 4: The correspondence between the tight binding Hamiltonian in Eq. (1) and Fig. 1(a) is not clear. I am not suggesting it is wrong, I am just saying that by looking at the Hamiltonian and the figure, it is difficult to relate the relevant terms to the corresponding sites. A better, enlarged figure with more appropriate description is needed.

Reply: We apologize for the confusion. To clarify it, we added an enlarged inset figure with detailed description in Fig. 1a. In addition, we highlighted the symbols 1-5 in the unit cell with red color (see also Fig. R4).

Fig. R4 Illustration of the original (left) and revised (right) Fig. 1a.

Comment 5: Similarly, it is difficult to understand the actual circuit structure from Fig. 2 (a). I could not understand what the dots represent and how to relate them to the actual electric elements.

Reply: We added a detailed description of Fig. 2a in the main text to clarify this issue: “The green and blue line segments represent the capacitors and , respectively, and each node is grounded by capacitors and inductors with the configurations shown in the insets.”

Comment 6: I could not understand how the features of the presented structure actually relate to the Weyl-semimetal.

Reply: We thank the reviewer for raising this question. We have added several sentences to describe the Weyl semimetal in the introduction: “For the conventional Weyl semimetal, the 3D topology features 2D gapped surface states that connect the projection of Weyl points on the surface [18-21]. Very recently, higher-order Weyl semimetal was reported which supports both the 2D surface Fermi arcs and the one-dimensional (1D) hinge state [22-29].”

As shown in Fig. R5, our results show the fingerprint features of higher-order Weyl semimetals, namely the Weyl points (hollow and solid circles with opposite topological charges) connected by the “Fermi arc” and hinge states in both real and momentum spaces.

Fig. R5 **a** “Fermi arc” of the surface states at 860 kHz. The color map and the white circle represent the experimental and theoretical results, respectively. The measured hinge-state voltage distribution in real space **b** and hinge-state dispersion in momentum space **c**.

Comment 7: Also, I could understand what is the big picture here? Why this work is important? Does it present any new physics? Does it have any interesting application? Does it provide a solution to any open problem? Does it provide more insight into a specific problem? Can it pave the way for exploring more interesting ideas? There is no discussion whatsoever about the impact or importance.

Reply: We thank the reviewer for arising these thought-provoking comments.

Big picture: Quantum mechanics is based on linear algebra, while the application of nonlinear operators can lead to outstanding discoveries under some circumstances, such as the prediction of positron, a direct result of Dirac equation which originally is derived from the Klein-Gordon equation by a square root operation. Recently, it has been shown that the square-root operator can be utilized to study topological insulators and to expand the horizon. Here, we, for the first time, generalize this approach to the gapless situations.

Importance: Since the discovery of the square-root topological insulator [Ref 4 Phys. Rev. B **95**, 165109 (2017)], the square-root operation has been intensively studied in gapped systems, ranging from conventional topological insulators to higher-order ones. However, it is still a mystery whether the square-root method can be applied to the gapless system. Our work fills in the cognition gap, evidenced by both the theoretical construction and experimental realization of the square-root higher-order topological Weyl semimetals.

New physics: We, for the first time, apply the nonlinear square-root operation to a three-dimensional gapless system and realize the square-root higher-order topological semimetal state. We use the simple inductor-capacitor circuit platform to observe the emergence of square-root higher-order Weyl semimetals with distinct Fermi-arc surface states and hinge states, which were not reported before.

Applications: We envision the following applications:

(1) Hinge states and surface states can be used to design robust solid-state devices. For example, one can transmit the signal with extremely low loss [Ref 46 Phys. Rev. Lett. **127**, 255501 (2021)], and devise the multiplexing [Ref 47 Phys. Rev. Applied **13**, 064031 (2020)] and imaging [Ref 48 Adv. Mater. **31**, 1904682 (2019)] with these topologically boundary modes.

(2) Topological LC circuit can be fully integrated with complementary metal-oxide-semiconductor (CMOS) technology [Ref 49 Nat. Nanotechnol. **17**, 262 (2022), Ref 50 Nat. Electron. **5**, 300–309 (2022)], which significantly reduces the circuit size and can be very useful for chip industry.

(3) Our work may spur the discovery of new materials with higher-order massless Weyl fermions, which can be utilized to conduct electric charge with faster velocity

[**Ref 17** Z. Phys. **56**, 330–352 (1929), **Ref 51** Nat. Phys. **11**, 645–649 (2015)]. Since Weyl particles are topologically protected from scattering [11,15], they could be very useful in quantum information science.

Open problem: We solved at least two open problems, considering the recent surge of interest in square-root topology and its generalizations (echoing Reviewer 2). Firstly, we theoretically construct the square-root operation on gapless system and provide a firm answer to the question if the square-root can be applied to metals besides insulators. Secondly, we experimentally realize the square-root higher-order Weyl semimetal in topoelectric circuits, thus provide solid evidence to our theoretical proposal.

Insight: The emergence of Weyl pairs in our work with both positive and negative energies mark its difference from the conventional higher-order Weyl semimetals. One of the parents sublattices, i.e., the honeycomb lattice, originally does not support any hinge states or flat-band states. The square-root operator makes it inherit these exotic states from the parent sublattice.

Exploring new ideas: Based on our findings, one may use the square-root method to explore many interesting new ideas for further study. For example:

(1) In electrical circuit, one may explore the enchanting phenomenon with square-root operations in higher-dimensional system, e.g., 4D [**Ref 52** Natl. Sci. Rev. **7**, 1288–1295].

(2) One may apply square-root operation to the Floquet topological insulators or semimetals, sustained by time-dependent periodic Hamiltonians [**Ref 50** Nat. Electronics **5**, 300–309 (2022), **Ref 54** arXiv:2111.14327 (2021)].

(3) Beyond the square-root, one can generalize the approach to the 2-root weak, Chern, and higher-order topological insulators, and 2-root topological semimetals [**Ref 15** Phys. Rev. B **103**, 235425 (2021), **Ref 11** Phys. Rev. B **104**, 165410 (2021)].

(4) Our work may inspire the exploration of the square-root higher-order topology in other solid-state systems, such as cold atoms, photonic crystals, and elastic lattices.

We have revised the manuscript to highlight these points.

Comment 8: In summary, I believe that the manuscript needs to be improved dramatically before it can be considered for publication in general. In addition, when it comes to publication in Nature Communications, the authors need to clarify what is the impact and significance of their work before it can be considered.

Reply: We thank the reviewer for the critical comments and useful suggestions. The manuscript has been revised carefully and the following discussions were added in the main text to clarify the impact and significance of our work:

“From the application perspective, both hinge states and surface states can be used to design robust solid-state devices. For example, one can transmit the signal with extremely low loss [46] and devise the multiplexing [47] and imaging [48] with these topologically boundary modes. Recently, it is shown that the topological LC circuit can be fully integrated by complementary metal–oxide–semiconductor (CMOS) technology [49,50], which significantly reduces the circuit size and can be very useful for chip industry. Our work may spur the pursuing of new materials hosting higher-order massless Weyl fermions, which can be utilized to conduct electric charge with faster velocity [17,51]. Since Weyl particles are topologically protected from scattering [11,15], they could be useful in quantum information science. Based on our findings, one may use the square-root method to explore many interesting new ideas for further study. For example, in electrical circuit, one may discover the enchanting phenomenon with square-root operations in higher dimensions, e.g., four-dimension (4D) [52]. One

may apply square-root operation to the Floquet TIs or semimetals, sustained by time-dependent periodic Hamiltonians [50,53]. Beyond the square-root, one can generalize the approach to the 2-root weak, Chern, and HOTIs, and 2-root topological semimetals [11,15].”

Response to Reviewer 2:

Comment 1: The authors study a square-root higher-order Weyl semimetal by generalizing to 3D in a specific way (namely one that breaks M_y symmetry, and therefore I -symmetry, for $t_z > 0$, even when $t_a = t_b$) the 2D model considered in their previous work (Ref. [6]), and implemented it in an electrical circuit. Due to the square-root nature of the model, the Weyl pairs appear at symmetric energies, which is a novel feature brought forth by this study. The experimental realization shows unambiguous signatures of the Weyl points, hinge states and Fermi arcs of the system. Considering the recent surge of interest in square-root topology and its generalizations, as well as in Weyl semimetals, I believe these results will appeal to a broad community.

Reply: We appreciate the reviewer for spending his/her time to very carefully review our manuscript and for evaluating our work as “novel” and “will appeal to a broad community”.

Comment 2: While I find these results interesting and view them in a favorable light, their presentation feels a bit rushed and insufficiently revised by the authors before submission and, as a consequence, there are several comments, listed below, that have to be adequately addressed first by the authors before I can recommend this paper for publication.

Reply: We thank the reviewer for judging our work “interesting” and “view in a flavor light” and for raising the very helpful comments. We are pleased to address them and have revised the manuscript accordingly.

Comment 3: A reference to Eq. (5) should be included, possibly Ref. [7] again. In lines 36-39, 94-96 and 107-110, the authors connect the topology of the hinge states to the quantized bulk polarization. However, it has been recently shown that, in itself, this quantization is not sufficient to affirm the topological nature of these states – see, e.g., npj Quantum Materials 5, 63 (2020) and the authors’ Ref. [10] (which should be corrected, since it was published in PRB), where the zero-energy corner states of the breathing kagome lattice (the parent block of the authors’ model at each k_z) are shown to stem from edge invariants and not bulk invariants. As such, the lines mentioned above should be less assertive when relating the topological nature of the hinge states to the bulk polarization, and also a reference to the results of Ref. [10] should be made. **Reply:** We thank the reviewer for pointing out our omission and we have corrected Ref. [10] and included Ref. [7] as a reference to Eq. (5). We also cited the npj Quantum Materials paper as Ref. [31]. Following the reviewer’s suggestion, we calculated the edge topological invariant based on the method of Refs. [10,31].

Fig. R6 **a** Dispersions of the slab geometry for $k = 1.66$. The red line denotes the surface state. **b** The topological edge invariant (Zak phase) as a function of k .

We put Fig. R6 into the Supp. Inf. as *Supplementary Fig. 3* and added the following description in the revised manuscript: “As a comparison, we also calculate the edge topological invariant following the method of [10,31]. We identify the same phase transition point at $\pm k$, as shown in *Supplementary Fig. 3b*.” In the revised supporting material, we added the calculation details as: “For the surface band in *Supplementary Fig. 3a*, the Zak phase is written as:

$$\mathbb{Z} = \int u^\dagger(k_x, k) V_{ku}(k_x, k) dk_x (\text{mod } 2\pi),$$

with $u(k_x, k)$ the wavefunction of the surface states. We calculate the edge topological invariant [2,3] and find the same phase transition point at $\pm k$ as shown in *Supplementary Fig. 3b*. Although it is not our purpose to resolve the debate between the edge invariant and bulk invariant to characterize the topology of hinge states, we sincerely show that these two approaches lead to the same prediction of the emergence of the hinge state in our lattice structure.”

Comment 4: In Fig. 2(a), it should be explained why the bottom line of nodes is included, since they belong to incomplete unit cells (the reason is that one requires a uniform onsite admittance for all nodes of the breathing kagome block that appears upon squaring the Laplacian). Also, the bottom edge line of blue nodes should be colored in dark gray instead.

Reply: We appreciate the reviewer for raising this issue. To clarify it, we added the following discussion to the caption of Fig. 2: “The finite-size equilateral triangular structure with the zigzag edge was chosen because one requires a uniform onsite admittance for all nodes of the breathing kagome block that appears upon squaring the Laplacian”.

The bottom edge line of nodes has been colored in dark gray. Thank you for correcting us.

Comment 5: In Fig. 2(b), while the hinge states are understandably 3-fold degenerate, I would expect the same to hold for the surface and bulk states, which appear to be instead 4-fold degenerate (or quasi-degenerate). Is there a simple reason for this?

Reply: We thank the reviewer for raising this issue. The hinge states indeed are 3-fold

degenerate due to the C_3 symmetry. However, the surface and bulk states do not demonstrate a 4-fold degenerate (or quasi-degenerate) after a careful inspection, as shown in Fig. R7.

Fig. R7 A part (with states number 1-50 and 110-160) of the bulk states and the surface states are displayed.

Comment 6: The parameters defined below the Laplacian in eq. (7) are not correctly defined: (i) $J_{\{0B\}}$ should read as $J_{\{0B\}}=iw(3C_B+6C_z)-1/(iwL_B)$ instead, (ii) similarly $J_{\{0C\}}$ should read as $J_{\{0C\}}=iw(C_A+C_B+2C_z)-1/(iwL_C)$, and (iii) J_B is simply $J_B=iwC_B$, with another parameter, let us call it $J_Z=iwC_Z$, connecting nodes at adjacent layers. Notice that all diagonal inductor terms, including in $J_{\{0A\}}$, should have a negative sign.

Reply: We thank the reviewer for pointing out our omission. In the revised manuscript, we have corrected these errors.

Comment 7: In lines 163-164, is there a way for the authors to specify (e.g. with arrows) in one of the plots where the nodes selected for the impedance measurements are located?

Reply: We accept the reviewer's suggestion outright. We have specified the points selected for the impedance measurements in Fig. 3c (see also Fig. R8).

Fig. R8 Revised Fig. 3c with node indexes.

Comment 8: Concerning Fig. 4(b), I understand that the frequency spectrum was found through the method of Sec. V of the Supp. Inf. However, given that the resonance condition occurs at $f_c=755$ kHz, for other values of “ f ” the model will not be exactly the same (there are varying onsite admittances at nodes A, B and C away from f_c).

Then $f=860$ kHz is considered in Figs. 4(c) and 4(d), such that it corresponds to a slightly different model. Even though I fully appreciate the relevance of the results of Fig. 4, is there an argument – based, e.g., on an adiabatic connection to the $f_c=755$ kHz case – from where one can say that qualitatively similar results are expected for this case?

Reply: We thank the reviewer for raising this important issue. The resonant frequency $f = 755$ kHz in frequency spectrum corresponds the $j = 0$ Ω^{-1} in the admittance spectrum. However, the Weyl points emerge at $j = \pm 0.004082$ Ω^{-1} [shown in Fig. 4a], which corresponds to the frequencies $f = 860$ kHz and $f = 1441$ kHz. We have revised manuscript to explain this issue: “*The Weyl points emerge at $j = \pm 0.004082$ (shown in Fig. 4a), which corresponds to the frequencies $f = 860$ and 1441 kHz in Fig. 4b. However, $f = 1441$ kHz deviates a lot from the working frequency of our selected electric components, because the component values suffer a drastic change (the values of electric components depend on the frequency). So, we only consider $f = 860$ kHz to display the complete “Fermi arc” (see Fig. 4d).*”

Comment 9: Still on Figs. 4(c) and 4(d), what is the origin of the central peak in Fig. 4(d)? Also, how is k_x determined since there are OBC along x ? It is not clear what the exact geometry used for the experimental results in Fig. 4(d) is, since it seems to assume PBC along x and z but OBC along y . The authors should explain this figure in more detail.

Reply: We thank the reviewer for raising this important issue. The central peak in Fig. 4(d) originates from the bulk state in the experimental measurements, which mixes with the surface state. k_x is determined as follows: “*Here, we determine k_x by $k = 21\pi m/N$, with $m = 1, 2, 3, \dots, N - 1$, and N being the number of grid points in the \hat{x} direction. The geometry used for the experimental measurement in Fig. 4d is a side face of the triple prism. We consider periodic boundary condition along \hat{x} direction and open boundary condition along both \hat{y} and \hat{z} directions.*” Details were also added in Methods section “*The geometry used for the experimental is a triangular prism shown in Fig. 5a. The geometry used for the theoretical calculations in Figs. 4a, b is also a triangular prism (open boundary condition in $k - k$ plane and periodic boundary condition along k_x direction) including 286 nodes. The geometry used for the theoretical results in Figs. 4c, d, Supplementary Fig. 3a, and Supplementary Fig. 4 is a slab (open boundary condition along k_x direction and periodic boundary condition along both \hat{y} and \hat{z} directions) containing 155 nodes.*”

Comment 10: In the Methods section, the specific components used should be identified, as well as their tolerances (is it 2% for all of them, was it measured?). **Reply:** We accept reviewer’s suggestion and listed a table in the Methods which identify the specific components in experiment. Indeed, we first measure the tolerance of all components, and then select a part of them with the same tolerance of 2% for experiments.

Comment 11: In lines 216-221, the authors relate the boundary modes of the residual block to those of the parent block of the squared Hamiltonian. I see these blocks on the same level, such that it might not be completely precise to say the boundary modes of the residual block are inherited from parent block. This should be more clearly restated, e.g., by citing Phys. Rev. Research 2, 033397 (2020) and saying, as in this reference, that the spectra of the blocks are necessarily degenerate, apart from extra zero-energy bands at the residual block, while explaining at the same time that the degenerate

boundary modes of the residual lattice are not topological but impurity states, stemming from onsite admittance offsets at the boundary nodes when the squaring operation is applied, as illustrated in Fig. 9(a) of Ref. [11].

Reply: We thank the reviewer for raising this important issue. In the revised manuscript, we added the following discussion to address this issue:

“When the square-root operation is applied, the spectra of the blocks are necessarily degenerate [Ref 33 Phys. Rev. Res. 2, 033397 (2020)] and the degenerate boundary modes of the honeycomb lattice are not topological but impurity states, stemming from onsite energy offsets at the boundary nodes [Ref 11 Phys. Rev. B 104, 165410 (2021)].”

Comment 12: When OBC are considered along all directions, are third-order corner states present, albeit possibly buried in the bulk continuum? Is there a simple strategy or modification to the model that could make these states appear at a gap?

Reply: We thank the reviewer for raising this important issue. To observe third-order corner states when OBC are considered along all directions in our structure, we need to adjust the inter layer t_z parameter to be breathing [Ref 45 J. Phys.: Condens. Matter 34 104001 (2022)]. In the revised manuscript, we added the following sentence:

“In the present model, by adjusting the inter layer hopping t_z to be breathing, third-order corner states will emerge [Ref 45 J. Phys.: Condens. Matter 34 104001 (2022)].”

Comment 13: In Figs. S1(c) and (d), should the caption not say instead something like “Berry curvature of the lower band of h_k^H for (c) $k_z=k_{zw}$ and (d) $k_y=\pm 2\pi/\sqrt{3}$ ”? Otherwise, it is not easy to understand what are the regions being considered, as they are not only around K_+ and K_- as indicated there and below Eq. (S10).

Reply: We appreciate the reviewer’s suggestion and changed it in the revised manuscript (see also Fig. R9).

Fig. R9 Revised Fig. 1 in Supp. Inf. The admittance dispersion around the Weyl points K_+ in the **a** $q - q_y$ and **b** $q - q_z$ planes. The spatial distribution of the Berry curvature **c** $k - k_y$ and **d** $k - k_z$ planes. Open and solid circles represent the Weyl points with opposite topological charges $+1$ and -1 .

Comment 14: In Fig. S2, the parameters in a-c should be swapped with those in d-f, since the $|t_a - t_b| < 2t_z$ has to be met for the Weyl points to appear. Also in Fig. S2(a), can the authors indicate to what surfaces the K and \bar{K} points belong to in the BZ? Comparing with Fig. 1(c), it would seem K is not simply $(4\pi/3, 0, 0)$, but belongs instead to a surface with $-\pi < k_z < k_{zw}$, while \bar{K} belongs to a symmetric $k_{zw} < k_z < \pi$.

Reply: We thank the reviewer for raising this important issue. In Supplementary Fig. 2, the parameters in **b-d** and **e-f** are revised in Supplementary Fig. 2a, we gave precise definitions about \tilde{K} ($\tilde{\Gamma}, \tilde{M}$) and \tilde{K} ($\tilde{\Gamma}, \tilde{M}$) correspond to the up and down translation of TI from \tilde{K} ($\tilde{\Gamma}, \tilde{M}$) along k_z direction, respectively (see Fig. R10a).

Fig. R10 Revised Fig. 2 in Supp. Inf. **a** The first Brillouin zone and the distribution of the Weyl points. The parameters in **b, c, and d** were chosen as $a = 0.5$, $b = 1$, and $c = 0.5$, corresponding to SHOWS. As a comparison, the parameters in **e, f and g** were chosen as $a = 0.5$, $b = 1$, and $c = 0.1$, corresponding to square-root HOTI. **b,e** Bulk band structures. Here, \tilde{K} ($\tilde{\Gamma}, \tilde{M}$) and \tilde{K} ($\tilde{\Gamma}, \tilde{M}$) correspond to the up and down translation of TI from \tilde{K} ($\tilde{\Gamma}, \tilde{M}$), respectively. **c,f** Bulk polarization P_1 as a function of k_x/π , with the subscript 1 indicating the 1st band. **d,g** The projected dispersion of a triangular prism, i.e., admittance along the k_z direction. The yellow line indicates the hinge state dispersion.

Comment 15: There are three papers in a pre-print version on square-root topology that I recommend the authors either cite as is or monitor and wait for their publication in the meantime before citing them:

arXiv:2108.01366 (2021)

arXiv:2109.00879 (2021)

arXiv:2111.14327 (2021)

Reply: We accept reviewer's suggestion and the papers were supplemented into our reference list.

[42] Rafi-Ul-Islam, S. M. Siu, Z. B. Sahin, H. Lee, C. H. & Jali, M. B. A. Unconventional node voltage accumulation in generalized topoelectrical circuits with multiple asymmetric couplings, arXiv:2108.01366 (2021).

[16] Kang, J. Liu, T. Yan, M. Yang, D. Huang, X. Wei, R. Qiu, J. Dong, G. Yang, Z. & Nori, F. Observation of Square-Root Higher-Order Topological States in Photonic Waveguide Arrays, arXiv:2109.00879 (2021).

[53] Bomantara, R. W. Square-root Floquet topological phases and time crystals, arXiv:2111.14327 (2021).

Comment 15: Misprints:

- Line 9: inherent → inherit
- Line 154: $L_A=33 \mu\text{H}$ → $L_A=30 \mu\text{H}$
- Below Eq. (S1): matrixs → matrices

Reply: We thank the reviewer for raising typos and correct them in the revised manuscript.

List of changes in the manuscript:

1. We have revised the manuscript carefully and polished the language.
2. We have added several sentences to describe Weyl semimetals in the introduction.
3. To clarify the correspondence between the tight-binding Hamiltonian in Eq. (1) and Fig. 1a, we added an enlarged inset figure with detailed description in Fig. 1a. We also highlighted the symbols 1-5 in the unit cell with red color.
4. To illustrate what the dots represent and how to relate them to the actual electric elements, we added a detailed description of Fig. 2a.
5. We calculated the edge topological invariant and added the Supplementary Fig. 3 in the supporting material. Meanwhile, we added discussion in both the main text and supporting material to explain this issue.
6. To explain why the bottom line of nodes in Fig. 2a are included, we added discussions to the revised manuscript.
7. We specified the nodes selected for the impedance measurements in Fig. 3c.
8. We explained why $\omega = 860 \text{ kHz}$ (rather than $\omega = 755 \text{ kHz}$ in Fig. 4d) was considered to measure the Fermi-arc in experiments.
9. We explained Fig. 4d in more details, including the method to determine .
10. We explained the sample geometries considered in Fig. 4, Supplementary Fig.2 and Supplementary Fig.3.
11. We added discussions in the Discussion and Conclusion section to highlight the impact and significance, and the potential application of our results.
12. In the Methods section, we added a table to list the specific component in experiment and stated clearly that we selected each component used in experiment with a tolerance 2%.
13. Twenty-five new references were added in the revised manuscript.
14. Typos were corrected in the revised manuscript.

REVIEWER COMMENTS

Reviewer #1 (Remarks to the Author):

Unfortunately, they authors have not addressed my comments successfully.

In particular, they have attempted to further explain the structure but without doing much effort. As a result, I still find it very difficult for any interested reader to reproduce their results. The additional inset in Fig. 1(a) does not help much. Also, the meaning of the dots and lines in Fig.2 (a) are not clear. If I would like to reproduce their results, how do I implement these dots and lines? The authors did not clarify this beyond the previous version.

Regarding the big picture, the authors argument is not convincing at all. The square root topology has been introduced already. The concept of Weyl semimetals is known. It is not clear what value the current work adds in the context of electronic circuits. Mapping the concepts from one domain to another by itself is not enough novelty. Also, the use of some complex terminology and complex sentences to enhance the presentation is not a good strategy. It just confuses the reader. As an example, what information does an unexpert reader get from the sentence: "the 3D topology features two dimensional (2D) gapped surface states that connect the projection of Weyl points on the surface"

In my opinion, this work lacks the clear presentation and novelty/impact to justify publication in Nature Communications

Reviewer #2 (Remarks to the Author):

The authors adequately addressed many of the issues raised by both referees and did some polishing on the presentation of the results, such that the quality of the manuscript has improved in this new iteration. While some issues with the language and phraseology persist, I believe some leeway should be granted to the authors on account of English not being their native language. I do, however, have some follow up comments that I would like see responded first before committing to a final decision.

(1) The parameters defined below eq. (7) were not completely corrected, namely with regards to the newly defined $J_{\{BZ\}}$ coupling. Since eq. (7) represents the real-space Laplacian, the nodes are connected either by C_B or C_Z , but not by both simultaneously. E.g., the entry $J_{\{12\}}$ should simply be $-J_B = -i\omega C_B$, since node 2 connects to node 1 with C_B only; it is the equivalent nodes to 2 on the top and bottom layers (one of them corresponding to node 288) that connect with $-J_Z = -i\omega C_Z$ to node 1. The authors must find a way to indicate in the Laplacian this J_Z coupling, which cannot be lumped together with J_B .

(2) The authors did not address my previous comment 5. In different terms, my question is: why are the bulk and surface states in Fig. 2(b), unlike the hinge states, not 3-fold degenerate, given the C_3 symmetry of the model? In my view, it would make sense that all states should be either 3-fold degenerate or non-degenerate (C_3 invariant states), and not only the hinge ones, but clearly Fig. 2(b) shows otherwise. What is, then, the flaw in my reasoning?

(3) The authors only partially answered my previous comment 8. As I understood it, the resonant frequency for which the diagonal terms of the Laplacian equally vanish is $f_c = 755$ kHz. Only for this frequency does one recover exactly the SHOWS proposed in Fig. 1; in other words, the frequency is locked at f_c for an exact mapping with the SHOWS. Then, a common grounding inductor $L_G = 22$ μ H is included to shift a hinge state to zero admittance. More importantly, if one changes the frequency from f_c , the diagonal terms yield different values, corresponding to alternating onsite admittances at

different types of nodes. The only way to access the Weyl points without changing the model would be to keep the $f_c=755$ kHz frequency fixed and change the L_G value. However, the authors do the converse and find a Weyl point at 860 kHz. Even though I agree with the authors' methodology, I think that a comment is perhaps in order here, since for $f=860$ kHz a slightly different model is obtained from the Laplacian. This is what I am referring to, the fact that it should be recognized and commented that a continuous variation of the frequency, in relation to f_c , also induces an adiabatic transformation of the model itself, when everything else is kept constant.

(4) Similarly, my previous comment 13 was not fully addressed. In Figs. S1 (c) and (d) the respective k_z and k_y values should be indicated, as well as the band for which the Berry curvature is being computed.

Misprints:

- On line 297 it is said that the triangular prism has 286 nodes. Is this a typo (that is, is it 2860 nodes) or is just one of the ten layers being considered here, with an implicit Fourier transform performed over the z direction?
- In the section III of the Supplementary Information and above eq. (S12), several references to the different plots of Fig. S2 indicate the wrong indices.
- Fig. S2b should be substituted by Fig. S2a of the previous version. Also, the authors forgot to include the sentence starting with "Here,..", present in the caption of Fig. R10 of their reply, in the caption of Fig. S2 itself.

Reply to reviewers' reports on Nature Communications manuscript NCOMMS-22-04759A by Song *et al.*

Response to Reviewer 1:

Comment 1: Unfortunately, they authors have not addressed my comments successfully. In particular, they have attempted to further explain the structure but without doing much effort. As a result, I still find it very difficult for any interested reader to reproduce their results. The additional inset in Fig. 1(a) does not help much. Also, the meaning of the dots and lines in Fig.2 (a) are not clear. If I would like to reproduce their results, how do I implement these dots and lines? The authors did not clarify this beyond the previous version.

Reply: We thank the reviewer to spend his/her time to review our work again. The correspondence between the tight-binding Hamiltonian Eq. (1) and Fig. 1a can be understood as follows:

As shown in Fig. R1, the nearest-neighbor sites of node 1(2) are nodes 3,4,5 with t_a , t_b being the intralayer hopping parameters. The next-nearest-neighbor sites of node 2 are nodes 2, 3 and 4 in the adjacent layer with t_z being the interlayer hopping parameter. The meaning of the dots and lines in Fig. 2a are similar to the atoms and hoppings of tight-binding ball-and-stick model in Fig. 1a.

Fig. R1 The Fig. 1a in main text. Illustration of an infinite 3D stacked honeycomb-kagome tight-binding model. The unit cell including five nodes is represented by the dashed black rhombus. The intralayer hoppings are t_a (green) and t_b (blue) in the $x - y$ plane, and the interlayer double-helix hopping is t_z (brown). Insets: Right view of the cell.

Fig. R2 **a** The partial finite-size circuit model. **b** Image of the partial circuit diagram designed by *Altium Designer*.

In electric circuits, the dots and lines represent nodes and capacitors, respectively. To further illustrate the process of designing this circuit, we display our method in Fig. R2. Figure R2a is a partial model for a single-layer, and Fig. R2b is the circuit diagram for designing a single-layer printed circuit board using *Altium Designer*. The colored dots and lines in Fig. R2a correspond to the different colored sites and capacitors in Fig. R2b. For convenience, we solder the interlayer capacitors on this circuit board as well and connect different layers of the circuit boards through flexible flat cable.

In the revised manuscript, we added more descriptions, and hope this step-by-step guidance will enable anyone interested in our work to easily reproduce our results.

Comment 2: Regarding the big picture, the authors argument is not convincing at all. The square root topology has been introduced already. The concept of Weyl semimetals is known. It is not clear what value the current work adds in the context of electronic circuits. Mapping the concepts from one domain to another by itself is not enough novelty. Also, the use of some complex terminology and complex sentences to enhance the presentation is not a good strategy. It just confuses the reader. As an example, what information does an unexpert reader get from the sentence: “the 3D topology features two dimensional (2D) gapped surface states that connect the projection of Weyl points

on the surface”

In my opinion, this work lacks the clear presentation and novelty/impact to justify publication in Nature Communications

Reply: We thank the reviewer for the comments. But we respectfully disagree.

First, our work is an interdisciplinary study that connects square-root topology and Weyl semimetals. It is known that interdisciplinarity involves the combination of two or more academic branches into a research project. It can create brand new ideas by thinking across domain boundaries.

Fruitful examples include the gauge symmetry that is a cornerstone of our fundamental description of quantum physics as encoded in the standard model of particle physics. By mapping the dynamics from particle physics to condensed matter physics, one can successfully simulate the gauge theory in ultracold atoms and optical superlattice, which are published in high-profile journals like Science [367, 1128-1130 (2020)] and Nature [587, 392–396 (2020)], respectively.

Second, the square-root topology and the higher-order Weyl semimetals indeed are two known concepts, but their combination yields a new topological state of matter. More important, the square-root topology was only applied to gapped/insulating systems in previous studies. In this work, we, for the first time propose and realize the square-root higher-order Weyl semimetals (SHOWS) allowing both the Fermi arc states and the dispersive hinge states. The novelty and impact of our work can be summarized as follows:

- (1) The discovery of hinge states and surface states simultaneously existing in SHOWS can be used to design robust solid-state devices.
- (2) The LC circuit realization of SHOWS can be fully integrated with complementary metal–oxide–semiconductor (CMOS) technology, which may solve the outstanding challenges met in the chip industry.
- (3) Our findings may stimulate the effort in the broad physics and materials community to look for real materials that support SHOWS, since we have provided a rather general framework to construct the Hamiltonian.

To summarize, we believe that our work represents an important advance of significance to topological physics and material sciences and to the emerging topological circuits.

The reviewer’s opinion that using complex terminology and sentences may cause confusion is certainly right. The sentence mentioned by the reviewer “the 3D topology features two dimensional (2D) gapped surface states that connect the projection of Weyl points on the surface” represents one of the key properties of Weyl semimetals, which is widely known in the community. In this description, we merely use standard terminologies in the study of topological states of matter. We improved the presentation and justification in the revised manuscript.

Response to Reviewer 2:

Comment 1: The authors adequately addressed many of the issues raised by both

referees and did some polishing on the presentation of the results, such that the quality of the manuscript has improved in this new iteration. While some issues with the language and phraseology persist, I believe some leeway should be granted to the authors on account of English not being their native language. I do, however, have some follow up comments that I would like see responded first before committing to a final decision.

Reply: We appreciate the reviewer for spending his/her time again to very carefully review our manuscript and for evaluating our manuscript as “the quality of the manuscript has improved in this new iteration”. We thank the reviewer for providing useful comments which helped us improve the manuscript. We are pleased to clarify all comments raised by our respected reviewer.

Comment 2: (1) The parameters defined below eq. (7) were not completely corrected, namely with regards to the newly defined $J_{\{BZ\}}$ coupling. Since eq. (7) represents the real-space Laplacian, the nodes are connected either by C_B or C_Z , but not by both simultaneously. E.g., the entry $J_{\{12\}}$ should simply be $-J_B = -i\omega C_B$, since node 2 connects to node 1 with C_B only; it is the equivalent nodes to 2 on the top and bottom layers (one of them corresponding to node 288) that connect with $-J_Z = -i\omega C_Z$ to node 1. The authors must find a way to indicate in the Laplacian this J_Z coupling, which cannot be lumped together with J_B .

Reply: We thank the reviewer for raising this important issue. In the revised manuscript, we have corrected these errors. The circuit Laplacian was rewritten as:

$$J(\omega) = \begin{pmatrix} J_{OB} & -J_B & 0 & \cdots & 0 & -J_Z & 0 & \cdots_1 \\ -J_B & J_{OC} & -J_A & \cdots & -J_Z & 0 & 0 & \cdots_2 \\ 0 & -J_A & J_{OA} & \cdots & 0 & 0 & 0 & \cdots_3 \\ \vdots & \vdots & \vdots & \ddots & \vdots & \vdots & \vdots & \ddots \\ 0 & -J_Z & 0 & \cdots & J_{OB} & -J_B & 0 & \cdots_{287} \\ -J_Z & 0 & 0 & \cdots & -J_B & J_{OC} & -J_A & \cdots_{288} \\ 0 & 0 & 0 & \cdots & 0 & -J_A & J_{OA} & \cdots_{289} \\ \vdots_1 & \vdots_2 & \vdots_3 & \vdots & \vdots_{287} & \vdots_{288} & \vdots_{289} & \ddots \end{pmatrix},$$

with $J_{OA} = 3i\omega C_A - \frac{1}{i\omega L_A}$, $J_{OB} = 3i\omega(C_B + 2C_Z) - \frac{1}{i\omega L_B}$, $J_{OC} = i\omega(C_A + C_B + 2C_Z) - \frac{1}{i\omega L_C}$, $J_A = i\omega C_A$, $J_B = i\omega C_B$, $J_Z = i\omega C_Z$ and the number subscript of the dots indicating the column/row numbers.

Comment 3: (2) The authors did not address my previous comment 5. In different terms, my question is: why are the bulk and surface states in Fig. 2(b), unlike the hinge states, not 3-fold degenerate, given the C_3 symmetry of the model? In my view, it would make sense that all states should be either 3-fold degenerate or non-degenerate (C_3 invariant states), and not only the hinge ones, but clearly Fig. 2(b) shows otherwise. What is, then, the flaw in my reasoning?

Reply: We apologize for not explaining it clearly in the last version. To completely clarify this issue, we discuss the symmetry constraint on the wave functions as follows: For the 3D stacked HK lattice in our model, it hosts the three-fold rotational symmetry C_3 . Therefore, we have the commutation relation: $[\hat{R}, \hat{H}] = 0$, with the three-fold

rotational operator $\hat{R} = \begin{pmatrix} 1 & 0 & 0 & 0 & 0 \\ 0 & e^{-ik \cdot \mathbf{a}_2} & 0 & 0 & 0 \\ 0 & 0 & 0 & 0 & 1 \\ 0 & 0 & 1 & 0 & 0 \\ 0 & 0 & 0 & 1 & 0 \end{pmatrix}$ (same as the U matrix in the main

text).

As a result, the eigenfunctions of \hat{H} can be chosen as eigenfunctions of \hat{R}

$$\psi_n^{(m)}\left(\theta + \frac{2\pi}{3}\right) = R\psi_n^{(m)}(\theta) = \exp(im\frac{2\pi}{3})\psi_n^{(m)}(\theta)$$

with $m = 0, 1, 2$, n is the energy level, θ the polar angle of any site, and $E_n^{(m)}$ being the corresponding eigenenergy.

I. Triple degenerate hinge states:

To explain the degeneracy of the HK lattice, we begin from the simple breathing kagome lattice in two dimensions. It is noted that the Hamiltonian of the kagome lattice H_K satisfies the generalized chiral symmetry: $\Gamma_3 H_i \Gamma_3^{-1} = H_{i+1}$ with $i = 0, 1$ and the

chiral operator $\Gamma_3 = \begin{pmatrix} 1 & 0 & 0 \\ 0 & e^{\frac{2\pi}{3}i} & 0 \\ 0 & 0 & e^{\frac{4\pi}{3}i} \end{pmatrix}$, and $H_0 + H_1 + H_2 = 0$. As reported in the

literatures [Nat. Mater. **18**, 113–120 (2019), Ref 39 Phys. Rev. Res. **2**, 022028(R) (2020)], the corner states are essentially pinned to the zero energy due to the generalized chiral symmetry. Let $\psi_n^{(0)}$ be the eigenfunction of $E_n = 0$. Due to the C_3 symmetry, $\psi_n^{(1)}$ and $\psi_n^{(2)}$ are also the wave functions of $E_n = 0$. As a consequence, the corner states are triple degenerate. By inserting the honeycomb lattice to the kagome one, we obtain the two-dimensional (2D) HK lattice [Ref 6 Nano Lett. **20**, 7566-7571 (2020)], where the triple degenerate corner states are shifted to finite energy due to the extra on-site potential induced by the honeycomb lattice.

The 3D stacked HK lattice are formed by hybridizing the 3D honeycomb lattice with 3D kagome lattice. Based on the above discussions, the hinge states (forming by a string of corner states in 2D case) should be triple degenerate.

II. Singlet or double-degenerate bulk states:

In addition, the circuit Laplacian satisfies $J^* = -J$ in the real space and $J^*(\mathbf{k}) = -J(-\mathbf{k})$ in the reciprocal space, which means that our circuit system holds the time reversal symmetry.

$$\mathcal{K}\psi_n^{(0)} = \psi_n^{(0)}, \quad \mathcal{K}\psi_n^{(1)} = \psi_n^{(2)},$$

where the time reversal operator \mathcal{K} is complex conjugate. So, $\psi_n^{(0)}$ is singlet, and $\psi_n^{(1)}, \psi_n^{(2)}$ form the Kramers doublets (double degenerate). One can therefore observe the single and double-degenerate bulk states.

To further verify our theoretical prediction, we take the 2D breathing SSH model with C_4 symmetry as another example, as shown in Fig. R3a. The eigenstates are plotted in Fig. R3b. It shows that the bulk state is either singlet or double degenerated due to the time-reversal symmetry, and only the corner modes are four-fold degenerate due to the generalized chiral symmetry in the C_4 system.

We added the following discussion to the main text “*One can see the three-fold degeneracy of the in-gap hinge states due to the C_3 symmetry and generalized chiral*

symmetry [39]. In addition, the time reversal symmetry dictates the bulk/surface states being singlet or double-degenerate (Kramers degeneracy)”.

Fig. R3 **a** The 2D SSH model with black and red segments indicating the two kinds of hopping terms $t_a = 0.5$ and $t_b = 1$. **b** Illustration of eigenstates with red, blue and black dots representing the corner, edge and bulk modes, respectively.

Comment 4: (3) The authors only partially answered my previous comment 8. As I understood it, the resonant frequency for which the diagonal terms of the Laplacian equally vanish is $f_c=755$ kHz. Only for this frequency does one recover exactly the SHOWS proposed in Fig. 1; in other words, the frequency is locked at f_c for an exact mapping with the SHOWS. Then, a common grounding inductor $L_G=22$ μH is included to shift a hinge state to zero admittance. More importantly, if one changes the frequency from f_c , the diagonal terms yield different values, corresponding to alternating onsite admittances at different types of nodes. The only way to access the Weyl points without changing the model would be to keep the $f_c=755$ kHz frequency fixed and change the L_G value. However, the authors do the converse and find a Weyl point at 860 kHz. Even though I agree with the authors’ methodology, I think that a comment is perhaps in order here, since for $f=860$ kHz a slightly different model is obtained from the Laplacian. This is what I am referring to, the fact that it should be recognized and commented that a continuous variation of the frequency, in relation to f_c , also induces an adiabatic transformation of the model itself, when everything else is kept constant.

Reply: We thank the reviewer for raising this important issue. The analysis of the reviewer is completely correct.

Indeed, one can shift the admittance of surface state $j_n = 0.004082 \Omega^{-1}$ to zero by changing L_G . This is what we have done in measuring the hinge state (we move the admittance of hinge states to zero by introducing $L_G = 22 \mu\text{H}$). To demonstrate another experimental technique, we, alternatively, tune the frequency to measure the Fermi arc surface state with the same circuit structure. As the reviewer correctly pointed out, a continuous variation of the frequency represents an adiabatic transformation of our model, with everything else kept constant. To illustrate this feature, we also measured the surface state at a different frequency $f = 835$ kHz, and we clearly observed the surface arc. In the revised manuscript, we added Fig. R4 as Supplementary Fig. 5 with discussions as follows:

“In theoretical calculations, one can conveniently set different admittance j_n and analyze the spectrum. For circuit experiments, we can shift an arbitrary admittance to

zero by adjusting the value of L_G . This is what we have done for measuring the hinge states. To demonstrate another experimental technique, we, alternatively, tune the frequency to measure the Fermi arc surface state with the same circuit structure”, “A continuous variation of the frequency can be seen as an adiabatic transformation of our model, with everything else kept constant”, and “We analyze the Fermi arcs at $f = 835$ kHz of Weyl points in frequency space. Supplementary Fig. 5a shows the Fermi arcs with the same frequency of Weyl points. These surface states are clearly gapped, as shown in Supplementary Figs. 5 b-f”.

We replaced Fig. 4d in the main text with the right panel of Fig. R5.

Fig. R4 Fermi arc and dispersions of the slab geometry in frequency space. **a** The contour of the surface states at the frequency of the Weyl points ($f = 835$ kHz). The open and solid circles denote the Weyl points with opposite topological charges. **b-f** Dispersions of the slab geometry along the k_x direction for different k_z ($k_z = 1.66, 1.78, 1.975, 2.19, \text{ and } 2.35$ for **b-f** respectively). The solid red line denotes the surface state dispersion and the dashed line shows the position of the frequency $f = 835$ kHz.

Fig. R5 Illustration of the $f = 860$ kHz (left) and $f = 835$ kHz (right) Fig. 4b.

Comment 5: (4) Similarly, my previous comment 13 was not fully addressed. In Figs. S1 (c) and (d) the respective k_z and k_y values should be indicated, as well as the band for which the Berry curvature is being computed.

Reply: We thank the reviewer for raising this important issue. In the revised manuscript, we have changed the caption of the Figs. S1 (c) and (d) as: “*Here we are particularly interested in the 1st band, because the Weyl points appear in the intersecting between the first and second bands. The spatial distribution of the Berry curvature for the 1st band around **c** $k_x - k_y$ ($k_z = k_{zw}$) and **d** $k_x - k_z$ ($k_y = 0$) planes*”.

Comment 6: Misprints:

• On line 297 it is said that the triangular prism has 286 nodes. Is this a typo (that is, is it 2860 nodes) or is just one of the ten layers being considered here, with an implicit Fourier transform performed over the z direction?

Reply: It is just one of the ten layers being considered here, with an implicit Fourier transform performed over the z direction. We have modified description in the revised version “*The geometry used for the theoretical calculations in Figs. 4a, b is also a triangular prism (open boundary condition in $k_x - k_y$ plane and periodic boundary condition along k_z direction) including 286 nodes in each layer*”.

• In the section III of the Supplementary Information and above eq. (S12), several references to the different plots of Fig. S2 indicate the wrong indices.

• Fig. S2b should be substituted by Fig. S2a of the previous version. Also, the authors forgot to include the sentence starting with “Here,..”, present in the caption of Fig. R10 of their reply, in the caption of Fig. S2 itself.

Reply: We thank the reviewer for raising the typos and we have corrected them in the revised manuscript. We modified the Fig. S2 indices and added the missing sentences “Here, \tilde{K} ($\tilde{\Gamma}, \tilde{M}$) and \bar{K} ($\bar{\Gamma}, \bar{M}$) correspond to the up and down translation of π from K (Γ, M), respectively” in the revised manuscript.

List of changes in the manuscript:

1. We added more explanations to the circuit model.
2. The expression of the circuit Laplacian was corrected.
3. Several sentences were added to describe the degeneracy of states in the main text.
4. Discussions were supplemented to show the feasibility of changing the frequency to measure the surface states.
5. Fig. R4 was added to Supplementary as Fig. S5.
6. Figure 4d was replaced by a new figure.
7. The captions of the Figs. S1 (c) and (d) were changed as: “*Here we are particularly interested in the 1st band, because the Weyl points appear in the intersecting between the first and second bands. The spatial distribution of the Berry curvature for the 1st band around **c** $k_x - k_y$ ($k_z = k_{zw}$) and **d** $k_x - k_z$ ($k_y = 0$) planes*”.
8. Justification of significance was improved.
9. Presentation was improved and typos were corrected.

Reviewer #2 (Remarks to the Author):

The authors have convincingly addressed my subsisting concerns (I just leave a minor comment below). I am now comfortable in recommending this paper for publication.

I do recognize that my previous comment on the degeneracy of bulk and surface states (also in light of the clear example of the 2D SSH model the authors provided) was, in hindsight, a poor one. Of course, bulk and surface states need not be 3-fold degenerate, contrary to the hinge states. E.g., bulk states are c_3 -symmetric states which, due to time-reversal (TR) symmetry, can be either non-degenerate (for a TR invariant k_z momentum) or doubly degenerate (at symmetric k_z momenta). However, the authors relate this double degeneracy to Kramers doublets; given that the SHOWS is a spinless fermionic model that is mapped into a classical system (the electrical circuit), the TR operator squares to 1 and Kramers theorem does not apply here. As such, the reference to "Kramers degeneracy" should be removed from line 168.

Caption of Fig. S1: "in the intersecting" -> "at the intersection"

Reply to reviewers' reports on Nature Communications manuscript NCOMMS-22-04759B by Song *et al.*

Response to Reviewer 2:

Comment 1:

The authors have convincingly addressed my subsisting concerns (I just leave a minor comment below). I am now comfortable in recommending this paper for publication.

Reply: We appreciate the reviewer for spending his/her time again to review our manuscript and for recommending our paper for publication.

Comment 2: I do recognize that my previous comment on the degeneracy of bulk and surface states (also in light of the clear example of the 2D SSH model the authors provided) was, in hindsight, a poor one. Of course, bulk and surface states need not be 3-fold degenerate, contrary to the hinge states. E.g., bulk states are c_3 -symmetric states which, due to time-reversal (TR) symmetry, can be either non-degenerate (for a TR invariant k_z momentum) or doubly degenerate (at symmetric k_z momenta). However, the authors relate this double degeneracy to Kramers doublets; given that the SHOWS is a spinless fermionic model that is mapped into a classical system (the electrical circuit), the TR operator squares to 1 and Kramers theorem does not apply here. As such, the reference to “Kramers degeneracy” should be removed from line 168.

Reply: We have deleted “Kramers degeneracy” from line 168.

Comment 3: Caption of Fig. S1: “in the intersecting” -> “at the intersection”

Reply: We thank the reviewer for pointing out the typo and we have corrected it in the revised manuscript.